# Analyses of the performance of the Ebola virus disease alert management system in South Sudan: August 2018 to November 2019

Olushayo Oluseun Olu[1]*, Richard Lako[2], Sudhir Bunga[3], Kibebu Berta[1], Matthew Kol[2], Patrick Otim Ramadan[1], Caroline Ryan[1], Ifeanyi Udenweze[1], Argata Guracha Guyo[1], Ishata Conteh[1], Qudsia Huda[1], Malick Gai[1], Dina Saulo[1], Heather Papowitz[1], Henry John Gray[1], Alex Chimbaru[1], Kencho Wangdi[1], Steven M. Grube[3], Beth Tippett Barr[3], Joseph Francis Wamala[1]

1 Ebola virus disease preparedness team, World Health Organization, Juba, Republic of South Sudan,
2 National Ebola virus disease Incident Management Team Ministry of Health, Republic of South Sudan,
3 United States Centers for Disease Control, Atlanta, Georgia, United States of America

* oluo@who.int

**Data Availability Statement:** All relevant data are within the manuscript and its Supporting Information files.

## Abstract

South Sudan implemented Ebola virus disease preparedness interventions aiming at preventing and rapidly containing any importation of the virus from the Democratic Republic of Congo starting from August 2018. One of these interventions was a surveillance system which included an Ebola alert management system. This study analyzed the performance of this system. A descriptive cross-sectional study of the Ebola virus disease alerts which were reported in South Sudan from August 2018 to November 2019 was conducted using both quantitative and qualitative methods. As of 30 November 2019, a total of 107 alerts had been detected in the country out of which 51 (47.7%) met the case definition and were investigated with blood samples collected for laboratory confirmation. Most (81%) of the investigated alerts were South Sudanese nationals. The alerts were identified by health workers (53.1%) at health facilities, at the community (20.4%) and by screeners at the points of entry (12.2%). Most of the investigated alerts were detected from the high-risk states of Gbudwe (46.9%), Jubek (16.3%) and Torit (10.2%). The investigated alerts commonly presented with fever, bleeding, headache and vomiting. The median timeliness for deployment of Rapid Response Team was less than one day and significantly different between the 6-month time periods (K-W = 7.7567; df = 2; p = 0.0024) from 2018 to 2019. Strengths of the alert management system included existence of a dedicated national alert hotline, case definition for alerts and rapid response teams while the weaknesses were occasional inability to access the alert toll-free hotline and lack of transport for deployment of the rapid response teams which often constrain quick response. This study demonstrates that the Ebola virus disease alert management system in South Sudan was fully functional despite the associated challenges and provides evidence to further improve Ebola preparedness in the country.

**Funding:** The author(s) received no specific funding for this work.

**Competing interests:** The authors have declared that no competing interests exist.

## Author summary

The Democratic Republic of Congo announced its tenth outbreak of the Ebola virus disease on 1ˢᵗ August 2018. As part of the preparedness measures to prevent and rapidly contain any importation of the virus, South Sudan, a neighbouring country to the Democratic Republic of Congo implemented a surveillance system which included an Ebola alert management system. We analyzed the performance of this system with a view to provide information to inform planning and allocation of resources to the other components of Ebola virus disease preparedness and to understand the key issues and challenges with the system. Our findings show that more than half of the reported alerts did not meet the case definition of the disease, alerts were mainly detected in the high-risk states, the commonest source of alert detection were from health facilities and the community and the most common symptoms presented by the alerts were fever, bleeding, headache, vomiting and weakness/fatigue. This study demonstrates that the Ebola virus disease alert management system in South Sudan was fully functional despite the associated challenges and provided evidence to further improve Ebola preparedness in the country. We recommend that the observed challenges should be urgently addressed.

## Background

On 1 August 2018, the Ministry of Health (MOH) of the Democratic Republic of Congo (DRC) officially declared an outbreak of Ebola Virus Disease (EVD) in its Eastern province of North Kivu [1]. This outbreak, the tenth experienced by DRC occurred shortly after another EVD outbreak which infected 54 persons out of which 33 died (Case Fatality Ratio, CFR: 61%) in the Equateur Province was declared over [2,3]. As of 30 November 2019, the current outbreak had reported 3304 cases and 2198 deaths with a CFR of 66.5% making it the second largest in the history of the disease [4]. A risk assessment conducted by the World Health Organization (WHO) identified nine countries that border DRC namely Angola, Burundi, Central Africa Republic, Republic of Congo, Republic of South Sudan, Rwanda, The Democratic Republic of Tanzania, Uganda and Zambia as being at risk for cross border transmission of the current DRC outbreak. Four of the nine countries, Burundi, Rwanda, South Sudan, and Uganda were classified as very high risk and intensive preparedness activities were recommended [5]. Two separate cross border transmission events occurred in Kasese district of western Uganda, that borders DRC's North Kivu Province on 11 June 2019 and 29 August 2019 respectively. This resulted in the notification of a total of 4 cases, all of whom died [6,7].

Based on the advice of the International Health Regulations (2005) Emergency Committee, the WHO Director General declared the EVD outbreak in DRC as a Public Health Emergency of International Concern (PHEIC) on 17ᵗʰ July 2019 [8]. This decision was predicated on ongoing risk assessments which showed the increased regional and global risk of spread of the outbreak and need for coordinated and intensified global efforts to prevent its further spread [9,10]. The committee recommended intensification of readiness to detect and manage cases through active surveillance including zero reporting from the neighboring countries [11].

South Sudan jointly reported the first ever EVD outbreak with the DRC in 1976 [12,13] and sits in one of the ecological zones of the disease in Africa. This and the recurrent outbreaks in neighboring DRC constitute a significant threat of an epidemic in the country which could undermine the country's health security, health system resilience and the ongoing revitalized peace process. Thus instituting an effective EVD preparedness and response system is imperative. The foregoing coupled with the close cultural, historical and family links, cross-border

trade and movement through an extensive and porous border with DRC and a pervasively weak health system in South Sudan, informed the country's decision to institute EVD preparedness measures aimed at preventing and rapidly containing any importation of EVD cases starting from August 2018. Based on the WHO consolidated EVD preparedness Checklist [14], interventions which were prioritized include the establishment of a national multisectoral EVD coordination taskforce, development of a scenario-based preparedness plan, strengthening of surveillance at points of entry into the country, community and in health facilities including establishment of an EVD alert management system, training and deployment of EVD dedicated Rapid Response Teams (RRT) and establishment of EVD diagnostic and confirmatory capacity at the National Public Health Laboratory (NPHL) [15].

EVD alert management is a critical component of the EVD surveillance strategy [16,17]. Data generated from such a system is a repository of information and evidence for timely detection, investigation and containment of potential cases. Such information is also useful for reviewing and refocusing ongoing preparedness and future response efforts, allocating resources and providing evidence-based information for planning, supervision, monitoring and evaluation of preparedness efforts [18]. While a number of reviews, monitoring and evaluation of the ongoing EVD preparedness interventions including two joint monitoring missions [19] and a mid-term review have been conducted in South Sudan, none have comprehensively reviewed the data generated from the EVD alert management system hence the need for this study. The study therefore analyzed the current functionality of and describes the information from the EVD alert management system in South Sudan.

The objectives of the study are two-fold, to better understand the epidemiological profile of EVD alerts in the country with a view to using the information to guide planning and resources allocation to the other components of EVD preparedness in the country and second is to understand the key issues and challenges with the EVD alert management system so as to further improve and expand public health preparedness for EVD and other epidemics in the country.

## Methods

### Study design and setting

We conducted a descriptive cross-sectional study of the EVD alerts which were reported in South Sudan from August 2018 to November 2019 using both quantitative and qualitative methods. Quantitative data was obtained from the national EVD alert management database and qualitative data from existing reports on EVD preparedness in the country.

South Sudan, the most recently independent country in Africa has a landmass of 619,745 km$^2$ and an estimated 2019 population of 12.6 million people. It is bordered by Uganda in the South, DRC in the South West, Central Africa Republic in the West, Sudan in the North, Ethiopia in the East and Kenya in the Southeast. It is subdivided into 80 administrative counties, 32 States and one Special Administrative Area with its administrative capital located in Juba. Four of the States, Tambura, Gbudwe, Maridi, and Yei River States have close cross border links with the DRC and were classified as very high-risk for cross border transmission of the ongoing EVD outbreak. Although Jubek, Torit and Wau States do not share direct borders with the DRC, they were also designated as high-risk because they host international airports and ground crossings which receive passengers from the DRC and the other priority one countries.

There are intense population movements between the two countries for economic and sociocultural reasons. The humanitarian and security context in the country particularly in the high-risk states pose one of the greatest challenges for effective implementation of EVD preparedness interventions including surveillance and alert management. A disrupted health

system as a result of several years of civil strife before and after independence resulted in inadequate human resources for health, infrastructure and a weak surveillance system. Ongoing skirmishes between opposing political factions, poor road infrastructure, and a difficult terrain worsened by a prolonged rainy season and severe flooding in 2019 limited access to many of the counties in the high-risk states.

Healthcare services are delivered at both the community and formal healthcare facility levels in the country. A community-based health programme called the Boma Health Initiative (BHI) which deploys community health workers is used to deliver preventive and curative health services such as immunization, distribution of bednets, health screening and education at the community level. At the formal level, health services are provided by a network of primary health care units, primary health care centres, county (general) and state referral hospitals which deliver primary, secondary and tertiary health services.

### The EVD alert management system

In line with WHO's EVD preparedness guidelines, the MOH South Sudan established a national EVD alert management system in the country in August 2018. Under this system, the country defined two EVD case definitions, one for the community and the standard routine surveillance case definition used at health facility level (Table 1). On the basis of these case definitions, screening and active searches for EVD cases are conducted at various points including border points of entry, communities and health facilities. Alerts which meet the suspected or probable case definition are verified and epidemiological and laboratory investigation are conducted.

At the national level, alerts from various sources are transmitted through a toll-free hotline (6666) to the Public Health Emergency Operations Centre (PHEOC) alert management team. The alert management team together with the PHEOC Manager verifies whether an alert meets the case definition or not. At the state level, alerts are received by the State surveillance officers who verify if it meets case definition with the support of the State MoH Director General, WHO, and other partner agencies. Alerts that do not meet the EVD case definition are discarded and spot reports are prepared and disseminated to all EVD stakeholders. The discarded alerts are investigated for other common epidemic prone diseases (Fig 1).

For alerts that meet the EVD case definition, a national or state RRT is activated and mobilized to investigate (Fig 1). An RRT typically comprises of seven members namely a public health or surveillance officer or epidemiologist who is also designated as the team lead, clinician, infection prevention and control officer, risk communication officer, laboratory technologist, data manager and a logistician. The RRT takes a detailed history of the alert, collects blood samples or oral swab (if the alert case is dead), list and provide risk communication to the contacts, family members, and the local community of the alert. Blood samples are collected in Ethylene diamine tetra acetic acid (EDTA) and viral inactivation buffer tubes, placed in triple packaging and transported to the NPHL in the capital, Juba either by air or road for preliminary testing by GeneXpert.

Between August 2018 and October 2019, the NPHL regularly sent the samples by air to the Uganda Virus Research Institute (UVRI) for confirmatory testing for EVD and other hemorrhagic fevers using Reverse Transcription Polymerase Chain Reaction (RT-PCR) test method. Capacity for RT-PCR testing of samples was established at the NPHL in October 2019 so both preliminary and confirmatory tests can be conducted in-country. Test results from NPHL and UVRI are entered into a Microsoft Excel database and sent to the PHEOC Manager who then communicates them to the national EVD Incident Manager. The test results are shared with EVD task force members at both the national and state levels and disseminated through the weekly EVD situation reports and during the bi-weekly EVD task force meetings.

**Table 1. Definition of community and health facility EVD alerts in South Sudan–August 2018 to October 2019.**

|  | Community level | Standard for routine surveillance at health facility level |
|---|---|---|
| **Case definition** | Sudden onset of fever with history of travel to an Ebola affected area. OR Any form of unexplained bleeding from any part of the body. OR Any sudden unexplained death. | **Suspect case** Sudden onset of fever (≥37.8˚C) and no response to treatment for usual causes of fever, and at least one of the following signs: · Bloody diarrhoea · Bleeding from gums · Bleeding into the skin (purpura) · Bleeding into eyes · Blood in the urine · Bleeding from the nose · Miscarriage (spontaneous abortion) · Any other form of unexplained bleeding OR Any individual who within the past 21 days has had a history of travel from Ebola-affected areas OR contact with a person with such travel history OR a history of contact with a suspect, probable or confirmed Ebola case AND (a) Sudden onset of fever (≥37.8˚C) OR at least three of the following symptoms: · A headache · Anorexia (loss of appetite) · Diarrhoea · Vomiting · Lethargy or fatigue · Stomach/abdominal pain · Body pains (muscle or joint pain) · Difficulty in breathing · Hiccups · A sore throat · Rash · Difficulty in swallowing OR (b) Unexplained bleeding (with or without fever): · Bloody diarrhoea · Bleeding from gums · Bleeding into the skin (purpura) · Bleeding into eyes · Blood in the urine · Bleeding from the nose · Miscarriage (spontaneous abortion) · Any other form of unexplained bleeding OR (c) Any sudden unexplained death |

Basic details of all alerts irrespective of whether they meet case definition or not are entered into a Microsoft Excel database. A case investigation form is completed for the alerts which meet the case definition. Hard copies of the forms are kept in a file at the PHEOC. The laboratory results of blood or swab tests are entered into the case investigation form as soon as they are received at the PHEOC. A suspected case is isolated in one of the four permanent EVD isolation units in (Juba, Nimule, Yambio, and Yei) or at temporary holding units for supportive treatment while the laboratory results are being awaited. The cases whose results are EVD negative are discharged from the isolation unit and referred for appropriate care while a confirmed case is transferred to an EVD treatment unit (Fig 1).

## Data collection and analysis

For this study, we reviewed, cleaned and conducted descriptive analyses on the Microsoft Excel database of all alerts recorded from August 2018 to November 2019 (S1 Data). We

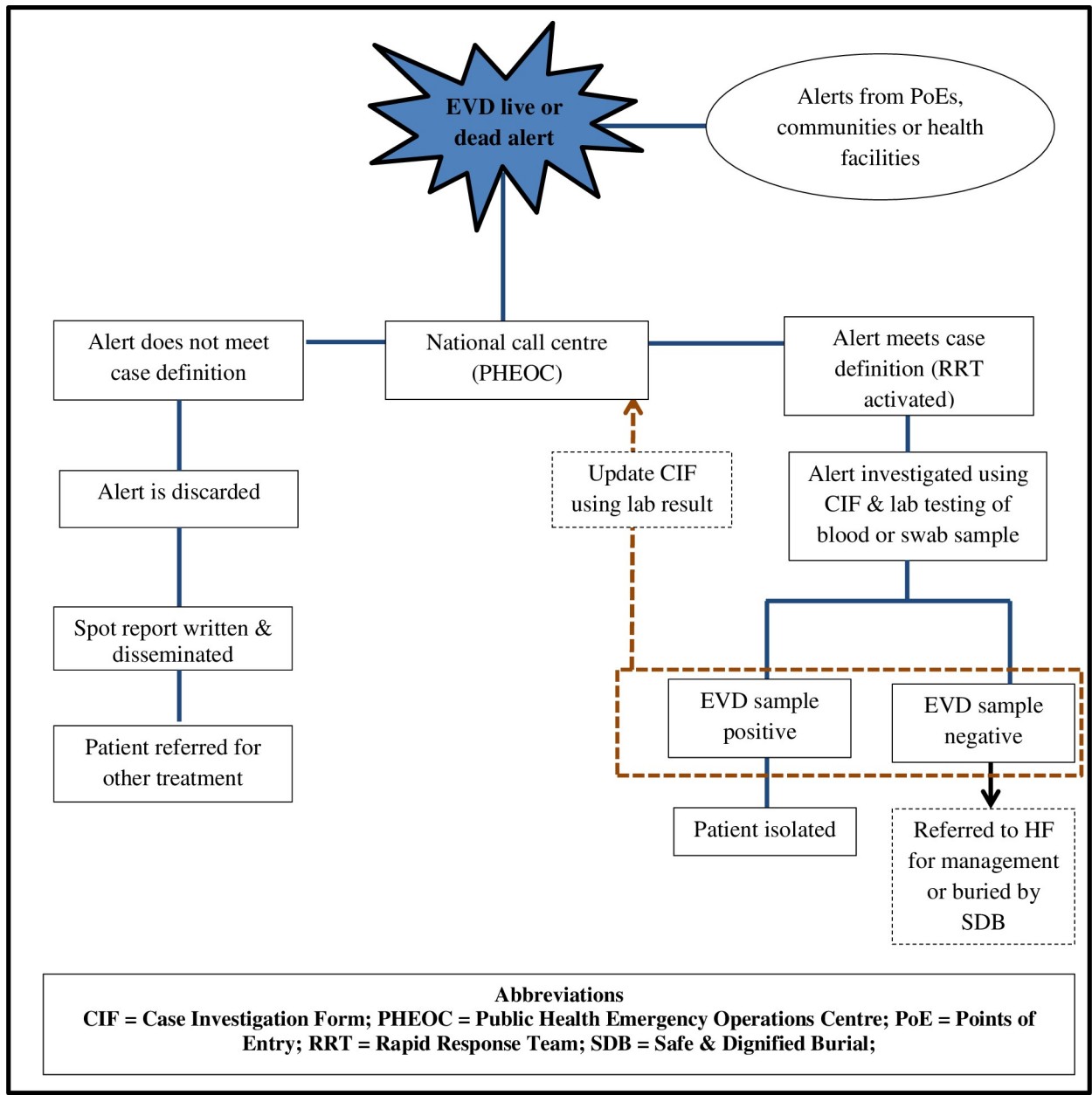

**Fig 1. Algorithm for Ebola Virus Disease alert management system in South Sudan–August 2018 to November 2019.**

retrieved and reviewed the case investigation forms of all the alerts that met case definition and the corresponding laboratory database, merged the relevant variables particularly the timelines for receiving and processing samples and results and entered them into the Microsoft Excel database. Other databases such as the Early Warning Alert and Response System (EWARS) that manages data from the Integrated Disease Surveillance and Response (IDSR) and the Early Warning, Alert and Response Network (EWARN) were also reviewed and all alerts that met the suspected EVD case definition were identified and included in the Microsoft Excel database. The IDSR system is the primary means of disease surveillance in the general population while the EWARN is used for disease surveillance in the internally displaced

people's camps. We cleaned and imported the database into Epi Info version 7.2.3.1 for windows where in-depth descriptive analyses were conducted.

For continuous variables like age, we computed the mean, median, mode, and standard deviation. On the other hand, for categorical variables like gender, source of alert identification, alert nationality, place of identification of the alert, alert symptoms, and alert epidemiologic links, we ran frequency distribution. Several indicators were computed to assess the performance of the EVD alert management system. The timeliness of alert notification was defined as the duration (in days) from onset of symptoms to the time the alert was picked up by the EVD alert network. Timeliness of RRT deployment was defined as the time (in days) from alert notification to actual deployment of the RRT to investigate the alert. Turnaround time (TAT) for alert investigation was defined as the duration (in days) when the alert was notified, and the PCR test result was received. TAT for GeneXpert EVD testing was defined as the time (in days) from receiving the EVD sample in the laboratory to the time the EVD GeneXpert test results were received by the EVD Incident Managers at national and state levels. TAT for RT-PCR EVD testing was defined as the time (in days) from receiving the EVD sample in the laboratory to the time the EVD RT-PCR test results were shared with EVD Incident Managers at national and state levels.

We analyzed the EVD alert performance data using the Anderson-Darling test of normality to assess the distribution of the EVD alert performance variables. The analyses showed that the p-values for timeliness of RRT deployment, TAT of alert investigation and RT-PCR test were all less than 0.05% indicating that the data on the EVD alert performance variables were not normally distributed. We therefore used the Kruskal-Wallis (K-W) test to assess if the changes for each of the timeliness variables between the six-month intervals were statistically significant.

For the qualitative aspect of the study, we used a strengths, weaknesses, opportunities and threats (SWOT) analyses methodology described by van Wijngaarden et al [20]. We extracted, compiled and reviewed information which are intrinsic to the EVD alert management system from the available EVD preparedness reports and documents such as the EVD joint monitoring missions, after action and mid-term reviews, EVD simulation exercises and EVD alert spot reports; while information external to the system was obtained from sources such as the South Sudan media monitoring reports, key context updates, economic outlook reports and South Sudan daily security situation reports. We then grouped the information into available resources or systems for EVD alert management (strengths and weaknesses) and stakeholders' expectations and contextual factors (opportunities and threats); a SWOT analysis matrix was then developed.

## Results

As of 30 November 2019, a total of 107 alerts had been reported in the country out of which 51 (47.7%) met the case definition and were investigated. Case investigation forms for 2 out of the 51 alerts could not be retrieved and were excluded from the analyses of symptoms and timeliness of investigation of the EVD alerts. The first alert was identified on 8 September 2018 while the last was identified on 23 November 2019. The incidence rate of EVD alerts was 0.93 per 100,000 population during the study period. Twenty-six (24.3%) of the investigated alerts were deaths (Fig 2). The mean, median and mode of the ages of the investigated alerts were 29.6 (standard deviation 17.4, confidence interval 14.5 to 21.7), 29 and 27 respectively (Table 2). Most of the investigated alerts were males and the most common sources of identification of the alerts were health workers at health facilities (53.1%), community (20.4%) and screeners at the points of entry (12.2%) (Fig 3). Majority of the alerts that did not meet the case

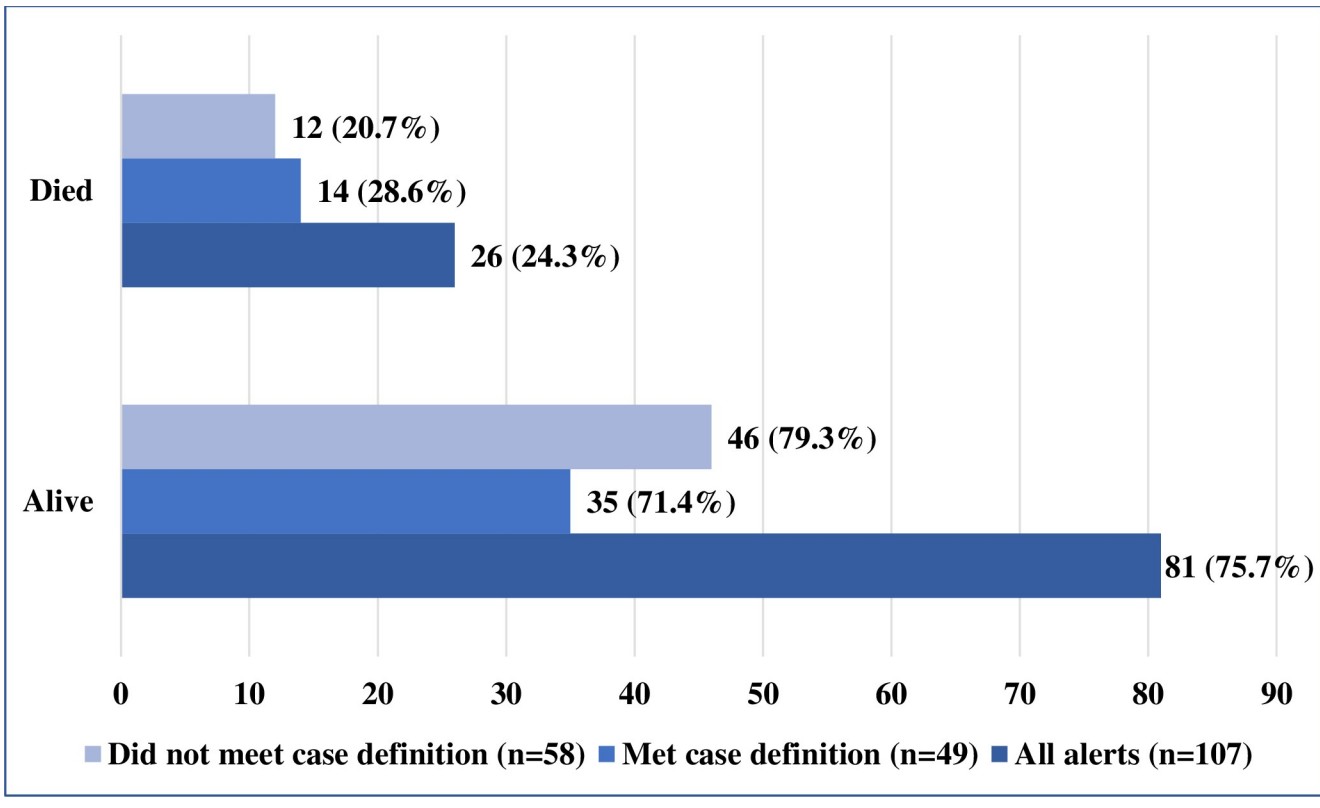

**Fig 2. Status of EVD alerts in South Sudan—August 2018 to November 2019.**

definition were identified from screening at the points of entry, health facilities, community and surveillance officers (Fig 3).

Most of the investigated alerts were South Sudanese (73.5%) and nationals of DRC (12.2%) (Table 2) who were mostly detected from the high-risk states of Gbudwe (46.9%), Jubek (16.3%) and Torit (10.2%) (Fig 4). Only 4 (8.2%) of the alerts were detected from the non-high-risk states (Fig 4). The highest number of alerts was detected in August 2019, followed by October 2019 and November 2018 (Fig 5). In August 2019, there was clustering of EVD alerts

**Table 2. Socio-demographic characteristics of EVD alerts in South Sudan–August 2018 to November 2019.**

| Variable | Category | No. of all Alerts (n = 107) | No. that met case definition (n = 49) | No. that did not meet case definition (n = 58) |
|---|---|---|---|---|
| Age | Mean | 27.2 | 29.6 | 25.2 |
| | Median | 27 | 29 | 22.5 |
| | Mode | 32 | 27 | 32 |
| Gender | Female | 31 (29%) | 11 (22.5%) | 20 (34.5%) |
| | Male | 76 (71%) | 38 (77.6%) | 38 (65.5%) |
| Nationality | Britain | 1 (0.9%) | 1 (2%) | 0 (0%) |
| | DRC | 7 (6.5%) | 6 (12.2%) | 2 (3.5%) |
| | Ethiopia | 1 (0.9%) | 1 (2%) | 0 (0%) |
| | Kenya | 3 (2.8%) | 2 (4.1%) | 1 (1.7%) |
| | South Sudan | 86 (80.4%) | 36 (73.5%) | 47 (81%) |
| | Uganda | 8 (7.5%) | 2 (4.1%) | 8 (13.8%) |
| | West Africa | 1 (0.9%) | 1 (2%) | 0 (0%) |

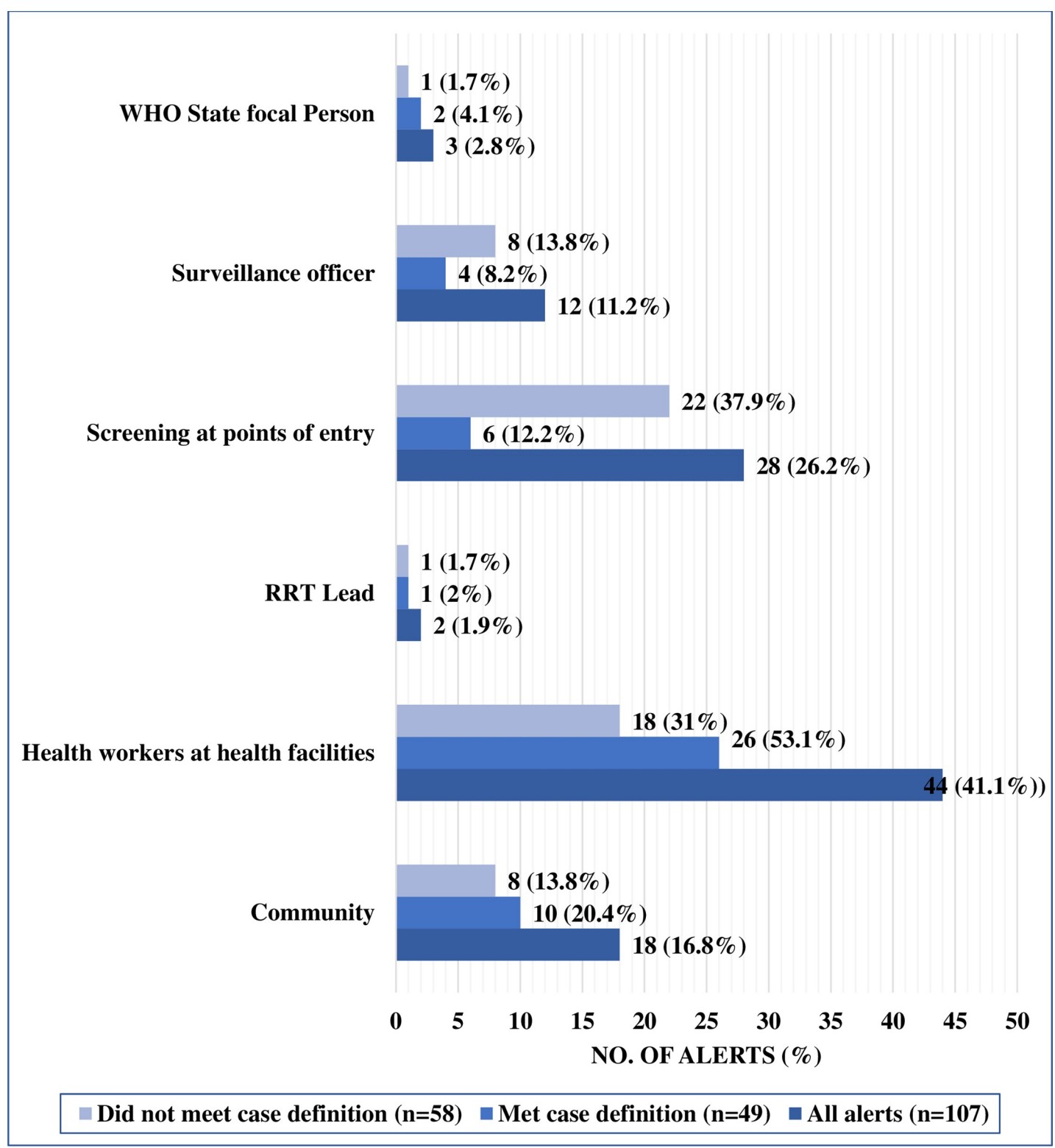

**Fig 3. Sources of identification of EVD alerts in South Sudan—August 2018 to November 2019.**

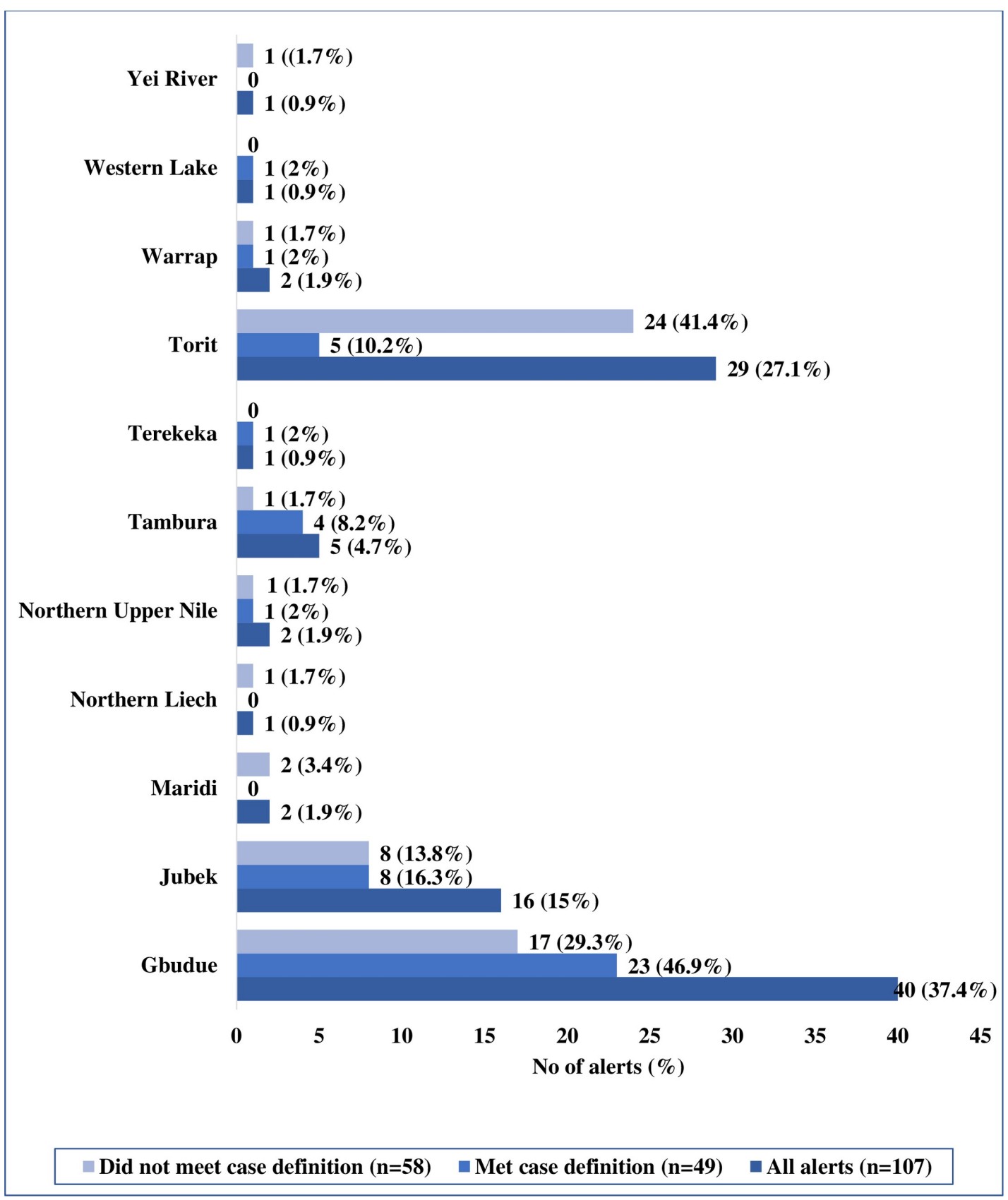

**Fig 4. Place of identification of EVD alerts in South Sudan—August 2018 to November 2019.**

in Yambio all of which were investigated and tested negative for EVD and other viral hemorrhagic fevers (Fig 6). The most common symptoms presented by investigated alerts were fever, bleeding, headache, vomiting, weakness/fatigue, loss of appetite and diarrhea (Table 3). Recent history of travel to EVD affected areas in the DRC was reported in 14.3% of cases. None of the suspected EVD cases reported history of contact with a confirmed EVD case.

The median timeliness of alert notification varied from 2.5 days in the second half of 2018 to 2 days in the second quarter of 2019. (K-W = 0.2719; df = 2; p<0.9). The median timeliness of RRT deployment was less than one day during the study period and significantly different between the 6-month time periods (K-W = 7.7567; df = 2; p<0.0024) while the median TAT for alert investigation improved from 2.5 days in the second half of 2018 to 1 day in the second half of 2019 (K-W = 20.382; df = 2; p = 0.000). The median of the GeneXpert TAT was 1 throughout the study period (K-W = 0.7662; df = 2; p<0.682). There was a significant improvement in PCR TAT from 2.5 days and 3.5 days in the second half of 2018 and first half of 2019 respectively to 2 days in the second half of 2019. (K-W = 16.711; df = 2; p<0.0002) (Table 4).

The strengths of the EVD alert management system during the study period included existence of a dedicated national alert hotline, case definition for alerts, rapid response teams and facilities for GeneXpert testing of samples at the national level. In addition, the existing IDSR and EWARN network of surveillance officers, RRTs and other resources provided a framework on which the EVD alert response capacities were built. The weaknesses identified with the system are among others, the dedicated alert toll-free hotline was often offline making reporting of alerts difficult, lack of transport to deploy the RRTs, inadequate documentation of alerts, incomplete members of RRTs and delays in collection and transportation of samples due to threats such as inaccessibility, unfavorable flight schedules and weather (Table 5). The report of the joint monitoring missions respectively showed two- and three-fold improvements in the capacity for EVD laboratory testing and epidemiological surveillance in the country from November 2018 to March 2019.

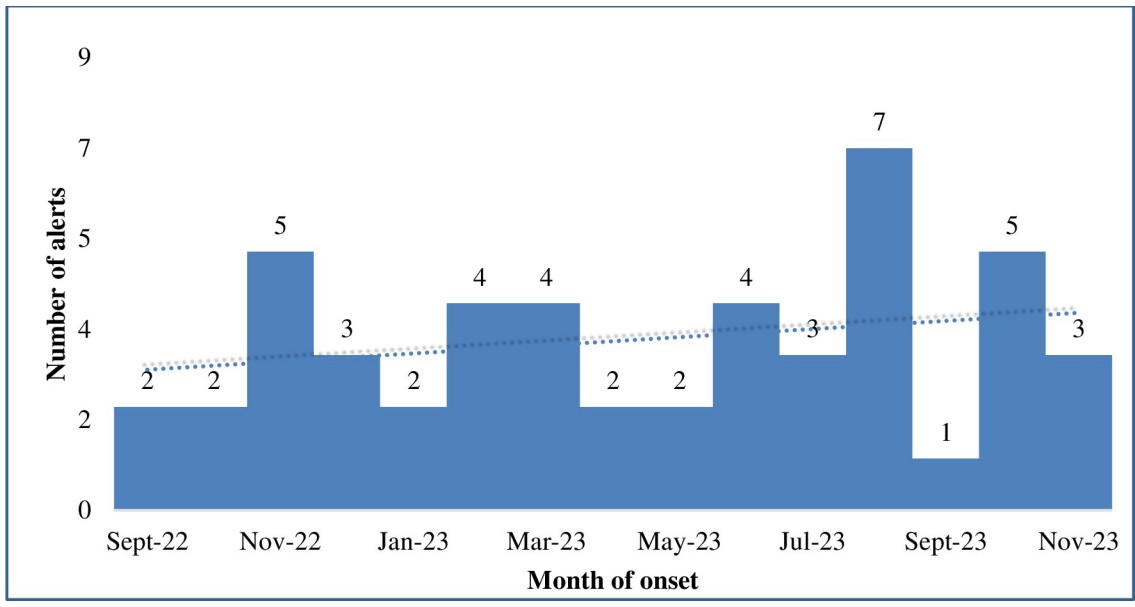

**Fig 5. Distribution of EVD alerts by months in South Sudan—August 2018 to November 2019.**

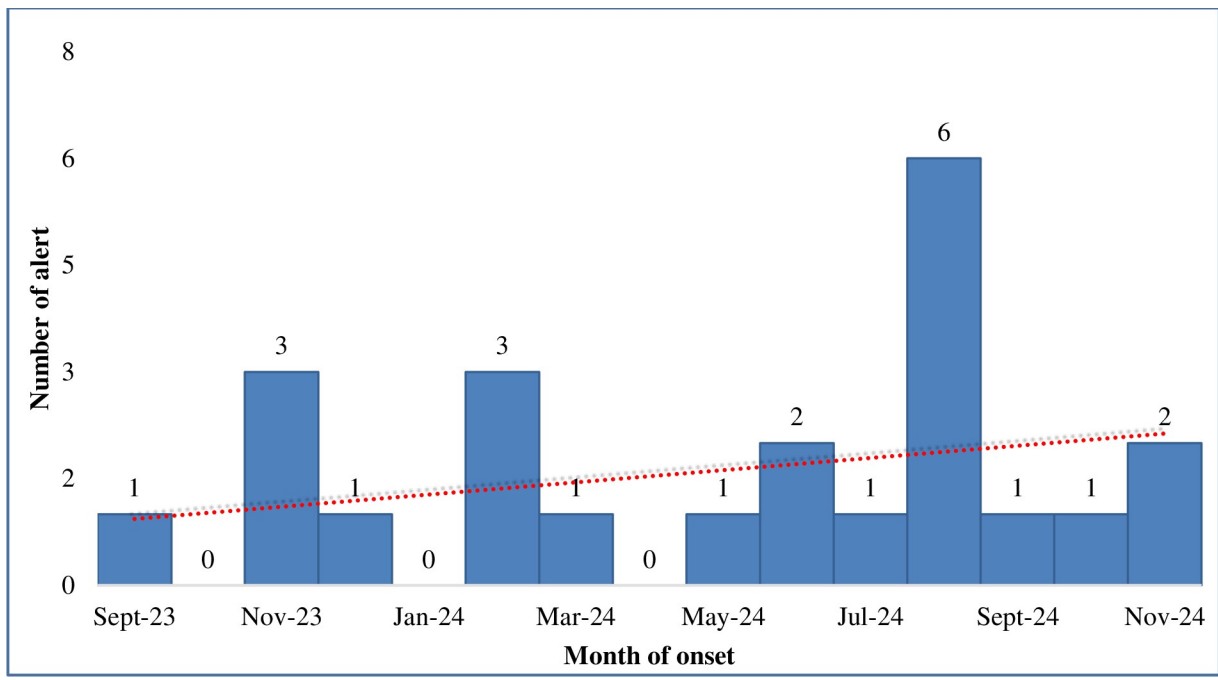

**Fig 6. Distribution of EVD alerts by months in Gbudwe State, South Sudan—August 2018 to November 2019.**

## Discussion

The importance of an EVD alert management system in preparedness for an outbreak cannot be overemphasized. This study sought to describe, characterize and draw lessons from the implementation of this critical system as part of preparedness for importation of EVD into South Sudan. The results show that more than half of the reported alerts did not meet the case definition of the disease, alerts were mainly detected in the high-risk states, the commonest

**Table 3. Symptoms associated with EVD alerts in South Sudan–August 2018 to November 2019 (n = 49).**

| Reported Symptom | No of alerts with symptom | Frequency of reporting (%) |
|---|---|---|
| Fever | 39 | 79.6% |
| Bleeding | 36 | 73.5% |
| Headache | 35 | 71.4% |
| Vomiting | 32 | 65.3% |
| Weakness/Fatigue | 29 | 59.2% |
| Loss of appetite | 24 | 49% |
| Diarrhea | 21 | 42.9% |
| Muscle pain | 20 | 40.8% |
| Joint pain | 18 | 36.7% |
| Cough | 17 | 34.7% |
| Chest pain | 11 | 22.4% |
| Dyspnea | 10 | 20.4% |
| Disorientation | 8 | 16.3% |
| Sore throat | 4 | 8.2% |
| Conjunctivitis | 3 | 6.1% |
| Hiccup | 2 | 4.1% |

**Table 4. Timeliness of investigation of EVD alerts in South Sudan–August 2018 to October 2019.**

| Variable | Period | No of observation | Total duration in days | Median | Statistical test of significance |
|---|---|---|---|---|---|
| **Timeliness of alert notification in days** | 2nd half of 2018 | 12 | 29 | 2.5 | K-W = 0.2719; df = 2; p<0.9 |
| | 1st half of 2019 | 18 | 56 | 2 | |
| | 2nd half of 2019 | 19 | 52 | 2 | |
| **Timeliness of RRT deployment in days** | 2nd half of 2018 | 12 | 9 | 0.5 | K-W = 7.756; df = 2; p<0.0024 |
| | 1st half of 2019 | 18 | 3 | 0 | |
| | 2nd half of 2019 | 19 | 2 | 0 | |
| **Turnaround time for alert investigation in days** | 2nd half of 2018 | 12 | 38 | 2.5 | K-W = 20.382; df = 2; p<0.000 |
| | 1st half of 2019 | 18 | 62 | 3 | |
| | 2nd half of 2019 | 19 | 29 | 1 | |
| **Turnaround time for GeneXpert lab test in days** | 2nd half of 2018 | 12 | 16 | 1 | K-W = 0.7662; df = 2; p = 0.682 |
| | 1st half of 2019 | 18 | 27 | 1 | |
| | 2nd half of 2019 | 19 | 20 | 1 | |
| **Turnaround time for PCR lab test in days** | 2nd half of 2018 | 12 | 42 | 2.5 | K-W = 16.711; df = 2; p<0.0002 |
| | 1st half of 2019 | 18 | 77 | 3.5 | |
| | 2nd half of 2019 | 19 | 37 | 2 | |

source of alert detection were from health facilities and the community and the most common symptoms presented by the alerts were fever, bleeding, headache, vomiting and weakness/fatigue. The study also demonstrated that most of the alerts that did not meet the case definition were detected from screening at the points of entry into the country.

## Incidence rate of EVD alerts

Compared to other high risk countries such as Uganda and Rwanda which had EVD alert incidence rate of 2.5 and 2.2 per 100,000 population (S1 Table) respectively, the EVD alerts incidence rate of 0.93 per 100,000 population in South Sudan is low. Given the very sensitive EVD case definition, population size and number of health facilities in the country, more alerts are expected. Furthermore, most of the alerts were clustered in the high-risk states, particularly in Gbudwe. This finding could be attributed to several reasons. First, at an estimated 28% [21] access to healthcare in the country is low and with no formalized countrywide community-based surveillance network, this could have prevented potential alerts from being detected at the community and health facility levels. Second, the pervasive insecurity in the country especially in the high-risk states may be a deterrent to present at health facilities. Third, the active component of the EVD alert system seems to have been focused mainly on the high-risk states and did not prioritize active search for alerts at the community and health facility levels especially in the non-high risk states. Fourth, EVD has been highly portrayed through the media as a disease that has no cure. This could be a deterring factor in a context where most health facilities do not have even basic resources to offer management of basic illnesses let alone EVD infection [22]. Additionally, EVD isolation and treatment units have been associated with confinement, mistrust and stigma in previous outbreaks [23,24]. Therefore, fear remains a deterring factor for individuals to present themselves at a formal health care facility, particularly if located far from their host communities. Perhaps, one of the most plausible reasons for the low number of alerts in the country is lack of awareness, good understanding and use of the case definitions by healthcare workers in the country. This reasoning is further buttressed by the fact that more than half of the reported alerts did not meet the case definition. However, the findings of this study demonstrate that the fact that more than half of the alerts did not meet the case definition is not necessarily a negative outcome as these cases were further

**Table 5. Strengths, weaknesses, opportunities and threats to the Ebola virus disease alert management system in South Sudan–August 2018 to October 2019.**

| Domain | Strengths | Weaknesses | Opportunities | Threats |
|---|---|---|---|---|
| Alert notification system | • Existence of a dedicated EVD alert hotline (6666) | • Inadequate number of staff to run the alert hotline 24 hours a day<br>• Inadequate number of dedicated lines<br>• Dedicated alert hotline is sometimes offline | • Availability of partners who are ready to support the running of the alert line<br>• Private mobile telephone companies were willing to support alert hotline through their corporate social responsibility mechanisms | • Unstable mobile telephone network in the country |
| Alert response and management system | • Existence of sensitive case definition for alerts<br>• Rapid response teams constituted and trained at the national and state levels<br>• Availability of isolation unit for management of alerts<br>• Availability of a national IDSR and EWARN network of surveillance officers, RRT and other resources (this provided backbone for EVD alert system) | • Lack of or inappropriate source of transportation for rapid response teams<br>• Inadequate documentation of alert investigation (incomplete or missing forms)<br>• Incomplete rapid response teams<br>• Rapid response team tool kit is heavy (70KG) sometimes making it difficult to transport<br>• Limited understanding of rapid response team on key guidelines and 72 hours response plan<br>• Inadequate understanding on the procedures and route for transmission of alert information<br>• High attrition rate of response team members due to lack of incentives | • Participation of the security forces in the national EVD taskforce (there is a security technical working group) | • Nationwide curfew which hampers alert response at certain times of the day |
| Alert confirmation system | • Existence of GenXpert machine and EVD cartridges at the national laboratory<br>• Availability of laboratory Technologists at the national public health laboratory<br>• Availability of sample collection kits at national and sub-national levels<br>• Possibility to confirm GenXpert results with PCR tests in neighbouring Uganda<br>• A PCR machine for Influenza vaccine was in the pipeline and could be be adapted for EVD testing | • Lack of capacity to confirm GenXpert results at the national level (this was addressed in October 2019)<br>• Delays in collection and transportation of samples from the field to the national level | • Availability of scheduled and chartered flights for sample transportation from sub-national to national level and to Uganda | • Widespread insecurity and bad roads which hampers access to the location of alert investigation<br>• Unfavorable weather conditions which sometimes affect flight schedules |

investigated for the benefit of the individual and to detect other potential diseases of public health concern.

## Epidemiological distribution of EVD alerts and their symptoms

Our findings demonstrate that health facilities and communities remain the primary sources of detecting alerts therefore the need for more effort to strengthen active surveillance at these levels. This finding is not surprising because most of the travelers who are screened at the points of entry screen are asymptomatic while the health facilities receive sick people who are more likely to have the same risk factors and non-specific symptoms included in the EVD case definitions. This observation supports the finding that most of the alerts that did not meet the case definition were identified from the points of entry and health facilities. This could be attributed to the level of awareness and understanding of the screeners at the points of entry and health workers on EVD symptoms and case definition. While these observations would

require further interrogation, they could guide allocation of resources to the various components of the EVD surveillance system in the future. The symptoms presented by EVD alerts are non-specific and similar to those observed in alerts and cases in other outbreaks [25,26]. However, a recent study documented discordance in the EVD case definitions which are being used for the current outbreak in DRC and the high-risk countries of Rwanda, South Sudan and Uganda; our findings thus provide evidence for ongoing evaluation and harmonization of these definitions as recommended by the study [27].

There was no proportionate increase in the detected number of alerts as the performance of the alert system improved over time which calls for a more active community mobilization and surveillance for alerts throughout the country. The clustering of EVD alerts in Gbudwe State in August 2019 coincided with deaths among cattle and triggered an outbreak investigation which detected East Coast Fever outbreak in the area. The EVD alert management system in South Sudan also detected an outbreak of Yellow Fever in Sakure, Nzara County of Gbudwe State in November 2018 [28]. This resulted in the implementation of a swift response vaccination campaign for which 95% of the targeted population were reached with the Yellow Fever vaccination which broke the transmission of the disease. This demonstrates the added value of the EVD alert management system but more importantly the need for an integrated and unified approach to surveillance of epidemic prone diseases in the country using effective and timely IDSR and EWARN systems. A plausible explanation for the reporting of the highest number of alerts from August to November of 2018 and 2019 is that this is the peak of the rainy season in South Sudan during which there is likelihood of higher incidence of other vector-borne tropical diseases such as malaria and Yellow Fever that share similar symptoms with EVD. Detailed epidemiological and laboratory investigations were unable to reveal the actual cause of the clustering of cases in Yambio in August 2019. Perhaps, this could be attributed to increased supervision and active search for EVD alert in Yambio as a result of the East Coast Fever outbreak during that period.

## Performance of the EVD alert investigation system

No statistically significant improvement was observed in the duration from onset of symptoms to the time the alert was picked by the EVD alert network. This could be attributed to low level of awareness and understanding of EVD and its symptoms and reporting of the suspected cases among the general population coupled with weak community surveillance structures and stigmatization during the study period further buttressing the need to intensify risk communication, community engagement and surveillance nationwide. Although the timeliness of deployment of RRT to investigate alerts improved over time, the RRTs experienced several challenges which hindered their performance. First, there was a high attrition rate among the RRTs mainly due to the lack of monetary incentives. Second, mobilizing RRTs for alert investigation was often hindered by lack of transport and materials required to conduct the investigation. Although in the latter part of 2019, an RRT investigation kit system was introduced to ensure that the materials were pre-packed and available, at 70 kilogrammes the kits were too heavy and bulky to be easily transported. Prior to this, the RRTs often lacked the full complement of the required equipment and personnel which delayed alert investigation. Third, although several RRT training sessions were held by various organizations that manage RRTs, the training sessions were not standardized which resulted in variable level of skills and performance of the RRT members.

Not much improvement in the TAT of the GeneXpert testing was observed during the study perhaps due to the lead time required to transmit the result from the laboratory to the EVD taskforces especially at the state level, inadequate proficiency and quality assurance in the

laboratory. Improving this performance indicator would require decentralization of GeneX-pert testing to the sub-national level which was shown to work well in Liberia [29] and institut-ing a quality assurance system at the NPHL. However, the operational feasibility and added value of decentralizing testing during the EVD preparedness phase in a complex humanitarian setting like South Sudan would have to be demonstrated. The significant improvement observed in the alert investigation and PCR TAT in the second half of 2019 is attributed to the establishment of PCR testing facility at the NPHL in Juba in October 2019 which eliminated the need to send EVD samples to UVRI which was previously practiced in the last half of 2018 and first half of 2019. The RT-PCR machine was originally procured for pandemic Influenza surveillance but is currently used to support confirmatory testing for EVD, Marburg virus dis-ease and other arboviral diseases like Yellow Fever, Rift Valley Fever and now the 2019 novel coronavirus disease. This further highlights the importance of an integrated approach to sur-veillance and laboratory testing for epidemic-prone diseases.

## Challenges associated with EVD alert management

This study identified several challenges to the EVD alert management system which are similar to the findings of other studies [30]. We propose a systematic approach to address these chal-lenges; the alert hotline should be monitored 24 hours and never switched off to ensure that alerts are reported in a timely manner. Provision of incentives, dedicated transport, training and on-the-job supervision to RRT members and streamlining of RRT kits would improve the timeliness and effectiveness of alert investigations. Importantly, establishment of a real-time online database for prompt documentation of alerts and regular training of RRTs using a har-monized training curriculum would improve EVD alert data completeness and quality while integration of the EVD alert system into the broader IDSR and EWARN systems would reduce duplication and ensure detection of a higher number of alerts especially in the non-high risk states.

## Study limitations

The findings of this study should be interpreted within the context of some limitations. Many of the case investigation forms were incomplete. For instance, many of the forms did not have data on the geographic information points so the alerts could not be mapped. Furthermore, we were unable to retrieve two of the case investigation forms for the alerts which were investi-gated. The total number of alerts generated during the study period was small and may not be representative of the true situation in South Sudan. To address this limitation, we conducted analyses on all the alerts using the Kruskal-Wallis test and aggregated the data by 6 months period to ensure adequate samples for analyses of the performance indicators.

## Conclusion

This study demonstrated that despite many challenges including weak health system, inade-quate access to health care services in several counties and difficulties in conducting surveil-lance as a result of insecurity, difficult terrain and lack of transport, the EVD alert management system in South Sudan was fully functional throughout the study period and improved progressively. However, there is still room for improvement to ensure that the sys-tem can timely detect and investigate more alerts. Our findings provide evidence for improv-ing the EVD alert management system in South Sudan and for informed decision making for strengthening and prioritization of EVD preparedness interventions particularly surveillance during the current and future outbreaks.

## Recommendations

Based on our findings we propose five key recommendations to improve the EVD alert management system in South Sudan. First, it is important to prioritize and intensify active surveillance and investigation of EVD alerts at the community and health facility levels all over the country through the IDSR and EWARN system and within the framework of the BHI to ensure that all EVD alerts are detected and investigated. This would entail revision, regional harmonization and wider dissemination of EVD case definition charts to major health facilities, sensitization and support supervision of health workers on the use of the definition and its differentials for capturing other infectious diseases of public health concern and scaling up of active search for alerts through regular visits to health facilities and communities. Furthermore, it is critical to scale up risk communication, community sensitization and engagement on the symptoms of EVD and other viral hemorrhagic fevers relevant to South Sudan. Second, establishment of a quality assurance system for the NPHL and decentralization of GeneXpert testing to the sub-national level could improve TAT for preliminary classification of alerts. Third, it is important to address the challenges observed with the EVD alert management system such as lack of transport, incomplete RRTs and limited understanding of RRT guidelines which are impediments to smooth running of the EVD alert systems. Regular RRT drills and simulations to address the weaknesses and improve the performance of the RRTs are required in this regard. Fourth, general improvement of the EVD alert management system could be informed by lessons taken from other countries [31]. Furthermore, this could be achieved through better integration of the alert management system into the IDSR and EWARN systems to further strengthen and sustain its capacity to detect, report, investigate, and respond to other priority diseases and public health threats. Fifth, improvements in the capacity and functionality of the EVD toll-free hotline system such as making it operational round the clock, publicizing it among the general population and health workers and regularly reviewing and evaluating it are recommended.

## Supporting information

**S1 Data. Line list of Ebola virus disease alerts in South Sudan: August 2018 to November 2019.**
(XLSX)

**S1 Table. Ebola virus disease alerts from the Democratic Republic of Congo and neighbouring countries.**
(PPTX)

## Acknowledgments

The authors thank all staff of the PHEOC Juba for availing the filled EVD case investigation forms used for this study. The authors alone are responsible for the views expressed in this article, which do not necessarily represent the views, decisions, or policies of the institutions with which they are affiliated.

## Author Contributions

**Conceptualization:** Olushayo Oluseun Olu, Richard Lako, Sudhir Bunga, Kibebu Berta, Patrick Otim Ramadan, Joseph Francis Wamala.

**Data curation:** Olushayo Oluseun Olu, Richard Lako, Sudhir Bunga, Kibebu Berta, Ifeanyi Udenweze, Malick Gai, Dina Saulo, Heather Papowitz, Kencho Wangdi, Steven M. Grube, Beth Tippett Barr, Joseph Francis Wamala.

**Formal analysis:** Olushayo Oluseun Olu, Richard Lako, Sudhir Bunga, Kibebu Berta, Caroline Ryan, Malick Gai, Kencho Wangdi, Steven M. Grube, Beth Tippett Barr, Joseph Francis Wamala.

**Investigation:** Olushayo Oluseun Olu, Richard Lako, Sudhir Bunga, Kibebu Berta, Matthew Kol, Patrick Otim Ramadan, Caroline Ryan, Ifeanyi Udenweze, Ishata Conteh, Qudsia Huda, Dina Saulo, Alex Chimbaru, Joseph Francis Wamala.

**Methodology:** Olushayo Oluseun Olu, Sudhir Bunga, Kibebu Berta, Matthew Kol, Patrick Otim Ramadan, Caroline Ryan, Ifeanyi Udenweze, Argata Guracha Guyo, Ishata Conteh, Qudsia Huda, Dina Saulo, Heather Papowitz, Alex Chimbaru, Joseph Francis Wamala.

**Project administration:** Olushayo Oluseun Olu, Richard Lako, Kibebu Berta, Matthew Kol, Patrick Otim Ramadan, Ifeanyi Udenweze, Argata Guracha Guyo, Henry John Gray.

**Resources:** Sudhir Bunga, Patrick Otim Ramadan, Argata Guracha Guyo, Ishata Conteh, Qudsia Huda, Malick Gai, Henry John Gray, Alex Chimbaru, Kencho Wangdi, Beth Tippett Barr.

**Software:** Olushayo Oluseun Olu, Kibebu Berta, Malick Gai, Kencho Wangdi, Steven M. Grube, Joseph Francis Wamala.

**Supervision:** Olushayo Oluseun Olu, Richard Lako, Kibebu Berta, Matthew Kol, Patrick Otim Ramadan, Caroline Ryan, Ifeanyi Udenweze, Argata Guracha Guyo, Dina Saulo, Heather Papowitz, Henry John Gray, Alex Chimbaru, Beth Tippett Barr, Joseph Francis Wamala.

**Validation:** Olushayo Oluseun Olu, Richard Lako, Sudhir Bunga, Kibebu Berta, Matthew Kol, Ifeanyi Udenweze, Ishata Conteh, Qudsia Huda, Heather Papowitz, Henry John Gray, Steven M. Grube, Beth Tippett Barr, Joseph Francis Wamala.

**Visualization:** Olushayo Oluseun Olu, Sudhir Bunga, Kibebu Berta, Matthew Kol, Patrick Otim Ramadan, Caroline Ryan, Argata Guracha Guyo, Ishata Conteh, Qudsia Huda, Malick Gai, Dina Saulo, Heather Papowitz, Henry John Gray, Alex Chimbaru, Kencho Wangdi, Steven M. Grube, Beth Tippett Barr.

**Writing – original draft:** Olushayo Oluseun Olu, Kibebu Berta, Joseph Francis Wamala.

**Writing – review & editing:** Olushayo Oluseun Olu, Richard Lako, Sudhir Bunga, Kibebu Berta, Matthew Kol, Patrick Otim Ramadan, Caroline Ryan, Ifeanyi Udenweze, Argata Guracha Guyo, Ishata Conteh, Qudsia Huda, Malick Gai, Dina Saulo, Heather Papowitz, Henry John Gray, Alex Chimbaru, Kencho Wangdi, Steven M. Grube, Beth Tippett Barr, Joseph Francis Wamala.

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
