## [Decision Letter · Decision Letter 0]

18 Aug 2020

Dear Dr Olu,

Thank you very much for submitting your manuscript "Analysis of the performance of the Ebola virus disease alert management system in South Sudan from August 2018 to November 2019" for consideration at PLOS Neglected Tropical Diseases. As with all papers reviewed by the journal, your manuscript was reviewed by members of the editorial board and by several independent reviewers. In light of the reviews (below this email), we would like to invite the resubmission of a significantly-revised version that takes into account the reviewers' comments. 

We cannot make any decision about publication until we have seen the revised manuscript and your response to the reviewers' comments. Your revised manuscript is also likely to be sent to reviewers for further evaluation.

Sincerely,

Nicholas P. Day

Associate Editor

Scott Halstead

Deputy Editor

Reviewer's Responses to Questions

**Key Review Criteria Required for Acceptance?**

**Methods**

-Are the objectives of the study clearly articulated with a clear testable hypothesis stated?

-Is the study design appropriate to address the stated objectives?

-Is the population clearly described and appropriate for the hypothesis being tested?

-Is the sample size sufficient to ensure adequate power to address the hypothesis being tested?

-Were correct statistical analysis used to support conclusions?

-Are there concerns about ethical or regulatory requirements being met?

Reviewer #1: Line 258: Methodological aspects for qualitative data analysis is inadequate. How were these data analysed? Did you use any framework? How did you synthesize the data from the review of documents? Please explain in detail. You can check COREQ guidelines for complete qualitative data reporting. Link: https://academic.oup.com/intqhc/article/19/6/349/1791966

Reviewer #2: The objectives are clearly articulated with a stated testable hypothesis. The study design appropriately addresses the stated objectives. Yes, the population is clearly described and appropriate. There is a limitation (also identified by the authors) in relation to sample size, but given there is no control on the number of possible EVD cases, the number of cases is adequate for a descriptive study. The authors applied the correct statistical analysis to support the conclusions. I find no concerns about ethical or regulatory requirements

**Results**

-Does the analysis presented match the analysis plan?

-Are the results clearly and completely presented?

-Are the figures (Tables, Images) of sufficient quality for clarity?

Reviewer #1: Page 16: since your study is focused on descriptive epidemiology of the surveillance system for Ebola. I wonder, if you could present some of these important variables in figures that would best serve the objective of your study. For e.g. 

• source of identification of alert

• Place of identification of alert

• Status of alert

Table 4: 

I see that you are trying to compare the timelines of alert notification in days between different parts of the year. But as you earlier said, your data had a non-normal distribution, thus you used non-parametric test: Kruskal Wallis test. For non-parametric test, it is best to present median (not mean or mode) and show only p value on the side. 

Table 5: It looks like a very informative table, and I believe you have used the SWOT analysis. However, could you delineate each component, importantly, ‘opportunities’ as recommendations? That will be something interesting for other LMICs and surveillance system.

Reviewer #2: The analysis and the plan match the clearly presented results.

**Conclusions**

-Are the conclusions supported by the data presented?

-Are the limitations of analysis clearly described?

-Do the authors discuss how these data can be helpful to advance our understanding of the topic under study?

-Is public health relevance addressed?

Reviewer #1: In the conclusion, you have mentioned Boma Health initiative (BHI). I think this is the first time you have talked about BHI. I recommend authors to provide a background information about BHI in the introduction section for readers to adequately anticipate and understand what it is and how does it come into the context.

Reviewer #2: The conclusions support the data presented, with the limitations of analysis well described. The in-depth discussion provide understanding on the public health relevance of the study

**Editorial and Data Presentation Modifications?**

Reviewer #1: (No Response)

Reviewer #2: I have addressed and identified some issues that need clarification in the body of the manuscript (uploaded). Table 1 should be presented in a landscape format

**Summary and General Comments**

Reviewer #1: Thank you for the opportunity to review the paper by Olu et al. In general, the paper is well written with contextual details concentrated in discussion section. There are some sections that needs improvement. Below, I have a few specific recommendations. 

Line 197: revise and mention the capital city; is it Juba? Or were you referring somewhere else?

Line 246: Mention here who are EVD stakeholders?

Line 402: IDSR and EWARN: please explain these abbreviations

Discussion is very well written and authors have explained in detail the interpretation and its practical relevance for South Sudan’s health system strengthening. I think the discussion will benefit from providing sub-headings for readers to easily pick up on the findings, challenges/recommendations.

Explain why there are highest alerts in August, October and November?

Why was there clustering of EVD alerts in Yambio?

Reviewer #2: The manuscript makes important contributions to the detection, response control of cross-border transmission of infectious diseases. While highlighting the challenges especially in a resource constrained environment, offering practical solutions for improving response activities

PLOS authors have the option to publish the peer review history of their article (what does this mean?). If published, this will include your full peer review and any attached files.

Reviewer #1: Yes: Bipin Adhikari

Reviewer #2: Yes: OYEWALE TOMORI
---

## [Editor Report · Decision Letter 1]

10 Oct 2020

Dear Dr Olu,

We are pleased to inform you that your manuscript 'Analyses of the performance of the Ebola virus disease alert management system in South Sudan: August 2018 to November 2019' has been provisionally accepted for publication in PLOS Neglected Tropical Diseases.

Best regards,

Nicholas P. Day

Associate Editor

Scott Halstead

Deputy Editor

---

## [Editor Report · Acceptance letter]

12 Nov 2020

Dear Dr Olu,

We are delighted to inform you that your manuscript, "Analyses of the performance of the Ebola virus disease alert management system in South Sudan: August 2018 to November 2019," has been formally accepted for publication in PLOS Neglected Tropical Diseases.

Best regards,

Shaden Kamhawi

co-Editor-in-Chief

Paul Brindley

co-Editor-in-Chief
